# Applying Customer Journey Mapping in Social Marketing to Understand Salt-Related Behaviors in Cooking. A Case Study

**DOI:** 10.3390/ijerph182413262

**Published:** 2021-12-16

**Authors:** Erik Cateriano-Arévalo, Lorena Saavedra-Garcia, Vilarmina Ponce-Lucero, J. Jaime Miranda

**Affiliations:** 1CRONICAS Center of Excellence in Chronic Diseases, Universidad Peruana Cayetano Heredia, Lima 15074, Peru; erikcateriano@gmail.com (E.C.-A.); lorena.saavedrag@gmail.com (L.S.-G.); Jaime.Miranda@upch.pe (J.J.M.); 2School of Medicine, Universidad Peruana Cayetano Heredia, Lima 15102, Peru

**Keywords:** salt intake, behavior change, customer journey mapping, social marketing, hypertension

## Abstract

Worldwide, salt consumption exceeds the World Health Organization’s recommendation of a daily intake of 5 g. Customer journey mapping is a research method used in market research to understand customer behaviors and experiences and could be useful in social marketing as well. This study aimed to explore the potential of customer journey mapping to better understand salt-related behaviors performed during the preparation of household cooking. We tracked the journey of four women in their kitchens for approximately two hours to observe the preparation of lunch. Individual journey maps were created, one for each woman, that were composited into a single journey map. We found that customer journey mapping was a suitable research method to understand how food preparers made decisions around adding salt and artificial seasonings at each stage of the journey. In contrast to the interviewee’ responses, it was observed that the four women added salt and artificial seasonings consistently and incrementally with little control and without any standard measure. In this study, we demonstrate the utility of customer journey mapping in a novel context and nudge social marketers to include this tool in their repertory of research methods to understand human behavior.

## 1. Introduction

One medical condition that currently affects approximately 626 million women and 652 million men around the world is hypertension, and over 1 billion people with hypertension live in low-and middle-income regions [1]. Hypertension consists of the persistent rise of blood pressure levels that can increase the risk of cardiovascular diseases, including stroke, heart attack, kidney disease, and others [2]. Hypertension is caused by the combination of physiological, social and cultural factors, and one of its major contributors is high intake of sodium, a mineral present in salt [2].

Worldwide, population salt consumption is higher than the World Health Organization (WHO) recommendation of 5 g/day and in many countries is around double [2,3]. The primary source of sodium intake varies by region and country. In developed countries, the major source of salt in the diet comes from processed and ultra-processed foods (products manufactured with the use of salt, sugar, oil or other substances added to preserve or to make them more palatable) [4]. By contrast, in many low-and middle-income countries the primary source of salt is adding salt during cooking or at the table, also referred as discretionary sources. Thus, given these heterogeneous contexts, health initiatives should shape salt reduction strategies according to local dietary patterns [4].

The World Health Assembly adopted the global target of a 30% reduction in mean population intake of salt by 2025 [5]. To achieve this goal, in 2016 the WHO developed a multilevel evidence-based strategy including the use of social marketing [6]. Social marketing has been proven to be effective in reducing salt consumption at a population level [7,8,9]. However, given the barriers reported by consumers as the negative effect in taste and the differences in acceptance of salt-reduced products, more in-depth understanding of salt-related behaviors is needed to develop tailored behavior change interventions to drive people into action [10].

Social marketing relies on formative research to understand targeted audiences’ behaviors [11]. One research method that permits tracking individuals’ emotions, needs, requirements, disagreements and observing their behaviors systematically is customer journey mapping [12,13]. Customer journey mapping has been largely used in market research as a method to assess how a service is currently functioning and what is the customers’ viewpoint about an existing service [12,14,15]. Therefore, customer journey mapping is typically used prior to the redesign of a service. The map of the customer journey takes the form of a flow diagram with multiple stages and a chain of touchpoints. The term touchpoint refers to moments of interaction that exert influence upon key customer behaviors. Touchpoints are generated by service providers, salespeople, websites, printouts, machines, other customers, peer influences, environments, and others [12,16]. Depending on the type of service/product the relevance of touchpoints may change along the journey map [16]. Customer journey mapping is flexible in terms of the methods used for data collection, although qualitative research methods, including interviews, focus groups, ethnography, and diaries, are predominant in customer journey articles [15].

In social marketing, customer journey mapping has been used as a tool to understand the viewpoint and experiences of individuals regarding the use of social programs and public services. For instance, one study in the UK applied customer journey mapping as an advocacy tool for blind and slightly sighted people to identify needs around the use of public transport and retail services [14]. Another study in the United States used journey maps to identify perceived benefits and barriers of mothers users of public health services provided by a supplemental nutritional program [13]. However, the application of customer journey mapping in formative research studies to inform behavior change interventions remains limited. In this article, we aimed to explore the potential of customer journey mapping to better understand salt-related behaviors in household cooking by closely tracking four women during the preparation of lunch. The journey mappings were also an opportunity to observe the use of artificial seasonings, another source of sodium in diet.

## 2. Materials and Methods

We conducted a larger formative research study that sought to understand perceptions of parents about salt consumption in Lima, Peru, between 2017 and 2018. This study was carried out in Miraflores and Mi Peru, two urban districts with high and low socio-economic status respectively and involved a non-probability sample of 296 participants. The first part of this larger formative research study had a mixed-methods design and its findings were published [17]. The second part of this larger formative research study consisted of applying four customer journey mappings, whose findings are described in the present article.

For our customer journey mapping study, a purposive sampling technique was used to recruit 4 women from the 296 participants (2 from Miraflores and 2 from Mi Peru). They were invited through their children’s kindergartens with the assistance of principals and teachers. Formal invitations were sent to school principals and written informed consents, detailing aims, record, and photographs’ purpose, were obtained from participants before customer journey maps took place.

Once recruited, the four participants were contacted by phone to arrange the date and time for the customer journey mapping. We mapped the journey of the four participants during the preparation of the lunch, generally regarded as the main meal among Peruvians. Customer journey mapping was conducted in each participant’s kitchen for approximately two hours. They were asked to prepare lunch as they usually do at home and describe in detail the cooking process. Photographs and field notes were taken and records of informal conversations were taped to better illustrate procedures/actions during the cooking process. An observation guide with a similar structure to a standard recipe was used. This guide included the following topics to explore: the name of the dish, the list of ingredients, the steps for preparation, and suggestions to serve the meal. If it was necessary, researchers formulated additional questions during the cooking process to clarify specific occurrences. All participants’ responses were audiotaped and transcribed with their permission.

Transcripts were entered into the qualitative software ATLAS-ti 8 (Scientific Software Development GmGH, Berlin, Germany). An initial list of codes and categories was developed *a priori* based on the larger formative research codebook [17]. Then, transcripts were distributed randomly among the researchers, who coded the documents using the refined list of codes. Codified transcripts were double checked in a second round of revision. Lastly, the three researchers triangulated the analysis using the observation notes, the photographs and the transcripts of the audios recorded during the observations.

Individual journey experiences were portrayed in the form of four case studies that were analyzed and compared in conjunction. Findings are reported in two sections. The first section consists of thematic analysis, while the second section is the description and analysis of the customer journey mappings. For the qualitative thematic analysis, the following themes were identified (1) use of salt during cooking, (2) perception of salt consumption, (3) use of artificial seasonings, natural spices and herbs during cooking, and (4) who decide what to cook at home. Four customer journey mappings were created, capturing stages, sub-stages, activities and touchpoints before, during and after the cooking process. In our study, touchpoints were activities in which participants touched or referred to salt, artificial seasonings and natural spices and herbs along the cooking process. We composited the four customer journey mappings into a single map identifying similar salt-related touchpoints and activities amongst the four participants and then prioritizing those that were repeated by three or all participants.

## 3. Case Study Descriptions

Participant 1 is a woman from Miraflores, who lives with her husband, two daughters and an aunt who looks after her daughters when she is working. Her younger daughter is four years old and studies in a private kindergarten in the same district. She works full time as a medical visitor. She cooked lasagna with minced beef and mushrooms for the observation. She shopped for her groceries at the local market and supermarket.

Participant 2 is a woman originally from Piura (northern Peru) who currently resides in Miraflores with her husband, one son aged four and one daughter aged seven. Her son studies in a public kindergarten located in Miraflores. She works sewing clothes at home in the mornings. For the observation, she chose *locro de zapallo* (smashed pumpkin with potato, cream milk and cheese) and bought all the ingredients in the local market, except the pumpkin, purchased in the supermarket.

Participant 3 is a woman from Mi Peru, a district in El Callao, Lima, that lives with her husband, her son, her mother and her brother. Her son studies in a private kindergarten in the same district. She works part-time selling clothes in her mother’s store. She prepared *tallarines verdes* (spaghetti with spinach and basil sauce) for the observation and shopped for her groceries very early in the local market.

Participant 4 is a migrant woman from Ayacucho (Peru’s Andean highlands) who moved to Lima when she was young and now lives in Mi Peru with her husband, her adolescent daughter and her younger son. Her son studies in a public kindergarten in the same district. She works selling cosmetics to her neighbors. She cooked *chanfainita* (stew with cow lungs) for the observation and bought the ingredients very early in the local market.

## 4. Results

### 4.1. Thematic Analysis

#### 4.1.1. Use of Salt during Cooking

All participants added salt constantly and in an incremental manner along the cooking process. Four specific times to incorporate salt were identified: (1) at the very beginning of cooking to flavor vegetables and meat; (2) during the preparation of the *aderezo* (a mixture of chopped onions, garlic, salt, pepper and oil that is base for most meals in Peru), in which salt was added at least twice; (3) during the preparation of sauces and main dishes to enhance the flavor; and (4) at the end of the preparation to check the flavor.

In all the case studies, participants measured salt differently and used different utensils. Overall, participants used spoons, teaspoons, or both to assist them measure salt, however these utensils had different sizes and shapes that made it difficult to establish a single measure. The amount of salt added by participants varied along the cooking process. One to two teaspoons of salt were added to prepare the *aderezo*, a half of a spoon was used to season single ingredients, sauces and main dishes and a quarter of a teaspoon was used at the end. A pinch of salt was identified as another measure in all participants.

Participants also had different perceptions of their salt usage. Participant 4 mentioned that she always added salt at the end of the cooking process. *“I add the salt when it (the meal) is cooked, I add the salt, trying. (…) I always add it at the end, trying.” (Participant 4).* However, our observation showed that she added salt since the beginning of cooking. Participant 1 mentioned that the most important time to add salt is the preparation of the *aderezo*. Our observation confirms her statement, given that she added salt to her *aderezo* three times.

In terms of measuring salt, two participants mentioned that their knowledge about measuring salt drew upon their practice and experience in cooking. *“Then you calculate (the amounts of) salt. In food there are not many measures... And salt… it is not (something) that I normally measure, I try it.” (Participant 1).* However, it was visible that multiple measures of salt were used by them along the cooking process.

#### 4.1.2. Perception of Their Salt Consumption

Overall, participants perceived themselves and their family members as low-salt consumers, but they added salt and artificial seasonings multiple times when cooking. All participants used the expressions “little salt” and “a very little salt” when referring to their salt consumption. *“And here I add a little bit of salt, it will be an eighth of a teaspoon, I grab the salt many times with my hand.” (Participant 1).*

#### 4.1.3. Use of Artificial Seasonings, Natural Spices and Herbs in Cooking

All participants suggested a variety of artificial seasonings, natural spices and herbs used alone or in combination, particularly when preparing the *aderezo* and sauces. Three out of four participants used artificial seasonings, being monosodium glutamate the most used artificial seasoning at home, followed by *Doña Gusta* (powder flavor seasoning). Onions and garlic were the main ingredients in the *aderezo*, although one participant incorporated paste of two kinds of chilies, *ají amarillo* and *ají panca*. Tomato, white pepper, and *hierbabuena* were used in sauces.

Our observations also revealed that participants had other natural spices and artificial seasonings in their kitchens such as black pepper, rosemary, oregano, dill, curcuma, *Sibarita* (powder flavor seasoning) and cloves, although none of them were used during our study observations.

Enhancing flavor and adding color were reported by all participants as two reasons to use both artificial seasonings and natural spices. *“I try to use the ají panca as little as possible. Above all, I just do it to add a little bit of flavour; sometimes it is also good to give it color.” (Participant 4).* Saving time when cooking was mentioned as another reason to use artificial seasonings. *“And there are times that I am behind the schedule and I do not want to mince the garlic or add salt, but not always; I add that new packet (of artificial seasoning with salt) that was launched (in the market).” (Participant 3).*

#### 4.1.4. Who Decides What to Cook at Home

We included the category who decided what to cook at home since we wanted to identify who were the principal actors in the cooking process. The four participants reported being the ones who decide what to cook at home. However, two of them mentioned that sometimes they ask their husbands or children for their input on what dishes they would like to be prepared. *“No, (I decide) by myself and some days, for example, when it is Friday or the weekend, I do consult with my children and with my husband. But normally I’m the one who decides what to cook.” (Participant 2).*

Table 1 presents a summary of findings comparing the responses of the four participants.

### 4.2. Customer Journey Mapping with the Four Participants

We created four customer journey mappings, one per participant, with stages, sub-tages, touchpoints, and activities occurring before, during, and after food preparation. In our article, touchpoints refer to times when participants mentioned or touched salt, artificial seasonings and natural spices and herbs along the cooking process. We composited the four customer journey mappings into a single map, identifying common touchpoints and activities amongst participants and then prioritizing those that were repeated by three or four participants. Customer journey mapping representations typically include positive and negative emotions of the participant experiences, yet in this study we did not find negative emotions around the use of salt in cooking.

Figure 1 shows the composite customer journey mapping. At the top of the figure, the first horizontal section shows the stages of journey (1) pre-cooking, (2) during cooking, and (3) post-cooking. These stages were defined according to the experience agreed upon after analysis of the four customer journey mappings. Each stage is segmented in two sub-stages that better account for the set of activities performed by participants along the cooking process. In the middle section of the figure there is a chain of 13 touchpoints, forming a curve. In the figure, salt-contact touchpoints are presented with a green dot, while non-salt-contact touchpoints are presented with a red dot. Finally, the lower part of the figure presents the recurrent activities that participants performed in each of the stages.

#### 4.2.1. Pre-Cooking

Four touchpoints were identified in this stage. Two of them are related to the substage shopping for groceries and other two were related to checking the ingredients. Participants answered some of the researchers’ questions during this stage. All participants mentioned that they bought the ingredients at the market nearby their homes, although time and budget limitations made it hard to follow participants and capture their experiences during shopping. None of the participants made a shopping list, therefore the researchers had to list the ingredients, while participants were checking and selecting the foods to be used. One participant commented that she changed the chosen meal to be cooked for the observation for a cheaper option, after considering the prices of the ingredients during shopping. None of the participants mentioned salt when ingredients were listed. However, all participants indicated the name and brand of the artificial seasonings and natural spices they would use to enhance the flavor of their meals. This last action was identified as the first salt-contact touchpoint represented in a green dot in the Figure 1.

#### 4.2.2. During Cooking

Five touchpoints were identified in this stage. The first three correspond to the substage preparing the *aderezo*, while the last two are related to the sub-stage called cooking. Participants prepared the *aderezo* basically with salt, artificial seasonings, onions, garlic and oil and followed similar cooking procedures. Three recurrent actions were observed: participants seasoned the *aderezo* with salt up to three times, then added artificial seasonings (e.g., Ajinomoto) and later added natural spices to enhance the flavor. The preparation of the *aderezo* had three touchpoints, being the sub-stage with more touchpoints along the cooking process, and each of these touchpoints were repeated/performed by all the participants. Therefore, in Figure 1 preparing the *aderezo* reached the highest peak on the curve of 13 touchpoints. Afterwards, participants added the rest of the ingredients to the *aderezo*. The last sub-stage was labelled cooking since it is related to cooking the meal. In cooking sub-stage, two clear actions were observed, namely tasting the flavor of the meal and flavoring with salt.

#### 4.2.3. Post-Cooking

In this stage, two sub-stages were identified, namely tasting the food and serving the food. All participants tasted the flavor of the food and flavored it with artificial seasonings or natural spices. Preparation of the meal was completed when the pots were removed from the stove burners. Participants let the meals settle down for a few minutes before starting to wash the dishes and prepare the table to serve the meal. At least two of the participants re-tasted the flavor of the meal minutes before serving it and then added salt. None of the participants put a saltshaker on the table at the time of serving the meal.

## 5. Discussion

### 5.1. Customer Journey Mapping to Understand Salt-Related Behaviours in Cooking

Customer journey mapping is a form of ethnographic observation that consists of tracking a person in a real context within a timeframe to understand his/her behaviors and experiences. In our study we explored the potential of customer journey mapping to better understand salt-related behaviors in household cooking, focusing on the use of salt and artificial seasonings along the preparation of lunch

Adding salt in cooking is the principal source of sodium in low-and middle-income countries. This cooking behavior is shaped by many factors, including taste, culture, the influence of family members and perceived behavioral control [18,19]. We found that customer journey mapping was a suitable research method to unpack the complexity of salt-related behaviors that take place during the cooking process. In creating the journey map, we identified the many different stages/sub-stages that participants go through in the preparation of lunch and what are the salt-related activities that participants perform along the cooking process to achieve the preferred taste. The depiction of the touchpoints allowed us to understand how participants made decisions around adding salt and artificial seasonings at each stage/sub-stage of the journey. We also identified patterns of salt usage in the repetition of some touchpoints along the cooking process.

In the case of the present study, the four participants incorporated salt and artificial seasonings consistently and incrementally without any standard measure or cooking utensil. The participants followed a similar pattern of salt usage to prepare the *aderezo* that involved a group of two touchpoints: (1) flavoring with salt and (2) flavoring with artificial seasonings. This salt usage pattern indicates that the preparation of the *aderezo* was important for them to ensure the preferred taste of their lunch, compared to the rest of the sub-stages. The emphasis that the participants put on the preparation of the *aderezo* might explain why there was a variety of artificial seasonings available in their kitchens. The patterns of salt usage in the preparation of the *aderezo* may be linked to regional/national culinary traditions in Perú [20]. Cultural patterns in cooking may shape taste preference, as it has been pointed out in some studies [21].

This study also confirms the utility of customer journey mapping to identify barriers and enablers to promote desired behaviors [12,16]. The depiction of the salt-related touchpoints facilitated the identification of barriers to reduce salt intake in cooking and provided potential customized solutions. For participants of this study, salt is an indispensable ingredient used in combination with artificial seasonings to achieve the preferred taste. Taste is one of the determinants of food choices and has been reported as a barrier to reduce salt intake in other studies [18]. Through the customer journey we observed that natural spices and herbs were also used by participants to enhance the taste of their preparations. Flavoring with natural spices was one of the touchpoints reported at different stages/sub-stages of the journey. The good predisposition to use these ingredients in cooking can be an opportunity to replace salt and improve their diet. The use of natural spices and herbs in cooking to reduce salt preference, salt intake and blood pressure has been effective in some countries [22,23,24].

Importantly, in contrast to what participants self-reported in terms of their habits and cooking practices, customer journey mapping was effective to elucidate salt-related behaviors in a real context, many of them unraveling higher intensities of salt usage. For instance, our participants mentioned that they often add little salt or very little salt during the cooking process, although the chain of touchpoints displayed in our map revealed that they added salt and artificial seasonings several times with little control and without any standard measure.

### 5.2. Implications for a Social Marketing Strategy to Reduce Salt Intake in Cooking

Through this study we have shown that customer journey mapping is applicable to the field of social marketing to investigate and promote healthy eating behaviors, thanks to its capacity to unfold complex behaviors that are not easily perceived or captured by traditional research methods. Customer journey mapping is suitable for social marketers since it allows to “accompany” the individuals through the performance of a task to understand his/her behaviors and develop potential solutions from the view of the individuals.

We observed that for our four participants adding salt in household cooking occurred as an unconscious action in which they did not pay attention to the times and amount of salt they added. Behavior change interventions to reduce salt intake at the population level often rely on messages alerting food preparers about the dangers of consuming excessive salt [25]. These kind of health messages may not be effective in the context of our study. The state of unconsciousness that food preparers manifested when they added salt in household cooking is a potential barrier to reduce salt intake. In this scenario, social marketers might adopt creative strategies that lead food preparers to visualize the actual amount of salt they use in household cooking, such as a cooking demonstration where food preparers weight the salt they use.

Our participants did not use a single measure to add salt. The variety of measures was determined by multiple factors, such as the number of family members, the type of meal to be cooked or the experience in cooking. Having multiple measures of salt might complicate the establishment of salt-reduction indicators in a social marketing intervention to reduce salt intake. The introduction of a salt-restriction spoon may be an effective strategy to standardize salt measure and reduce salt-intake among food preparers. Previous studies have reported that salt-restriction spoons helped to reduce the daily salt intake in home cooking [26].

The identification of touchpoints along the cooking process may serve to better choose and frame activities and messages for a social marketing campaign to reduce salt intake. Flavoring with salt and flavoring with artificial seasonings were two relevant and repeated touchpoints along the cooking process that allowed food preparers to ensure the preferred taste of their meals. Given the importance of taste, social marketers should find ways to reduce salt intake, without altering the flavor of food. As mentioned, using natural spices and herbs may be an alternative.

Our findings are complementary to our mixed-methods study that informed the development of a social marketing strategy for the reduction of salt/sodium consumption in Latin America [27]. In addition, this study wants to contribute to a pressing call from social marketing discipline for incorporating novel approaches and research methods in formative research to better understand human behavior [28,29].

### 5.3. Limitations

There are important limitations in this study. The first limitation is that the sample was small and given the way participants were enrolled and informed about the purpose of the study, there is a possibility of a selection bias. However, our sample size was determined by pragmatic considerations [30], in order to test the utility of customer journey mapping in a novel context (i.e., the kitchen). In addition, in case studies there is not an ideal number of participants. Some authors suggest that a sample ranging from 4 to 10 participants works well [31]. Furthermore, the number of participants recruited and the number of journeys reported in previous empirical studies about customer journey mapping vary [15]. To our benefit, we managed to secure a sample of participants from different socioeconomic and cultural backgrounds, including migrants into the urban capital, providing an adequate range of experiences across very heterogeneous contexts.

A second limitation is that the four customer journey mappings were focused on the perspective of mothers and the preparation of one single mealtime, namely lunch. Mapping the journey of other family members and focusing on the preparation of other meals (i.e., the dinner) could yield different findings. However, we consider this a minor bias, given that lunch is the most consistent meal in Peruvian households and most people consume food at home in this country [32,33].

Another limitation is that participant feedback to review the map of the journey created after data collection was not conducted. The literature suggests that customer input is important to validate the accuracy of the depicted journey map [16]. Yet, this process was not possible to carry out in our study. We addressed issues associated with validity and reliability of qualitative research by using two forms of triangulation (1) the use of multiple research methods, including observations, photographs and note-taking and (2) the involvement of three independent views provided by three authors of this article in the data analysis.

## 6. Conclusions

In this study, customer journey mapping was employed to track four women during the preparation of lunch to better understand salt-related behaviors. Our study demonstrated that salt was perceived as an indispensable ingredient to provide preferred taste in meals, incorporated consistently along the cooking process. The use of salt in cooking was complemented with other sources of sodium, such as artificial seasonings, although the use of natural spices and herbs was noted. Furthermore, customer journey mapping provided a unique opportunity to visualize salt intake behaviors in situ and consequently contrast what food preparers say against what they do. The four participants self-reported that their salt consumption was low, but they had little control over the amount of salt they used in cooking. As shown in this study, customer journey mapping is an ideal tool to enhance and strengthen the process formative research by providing a deep understanding of complex behaviors, such as salt/sodium intake, that cannot easily be perceived through self-reported methods. Moreover, in contexts where food preparers add salt during cooking, such as households in many low-and middle-income countries, customer journey mapping can be helpful to account for the cultural specificity of human behavior.

## Figures and Tables

**Figure 1 ijerph-18-13262-f001:**
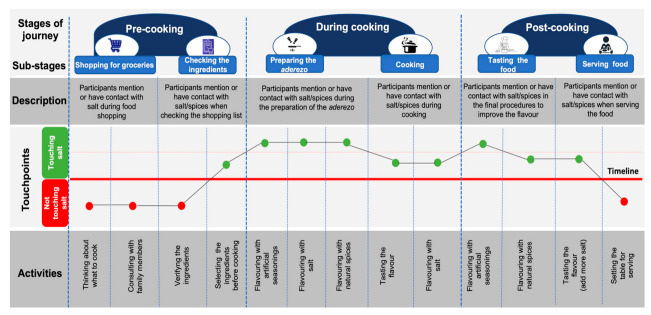
Participants’ composite customer journey mapping.

**Table 1 ijerph-18-13262-t001:** Participants’ cooking practices and experiences around salt and artificial seasonings.

Criteria	Participant 1	Participant 2	Participant 3	Participant 4
Times to incorporate salt	At the very beginning, during the preparation of the *aderezo*, during the preparation of the sauces, main dish and at the end to check the flavor	At the very beginning, during the preparation of the main dish and rice and at the end to check the flavor	At the very beginning, during the preparation of the sauce and spaghettis and at the end to check the flavor	At the very beginning, during the preparation of the *aderezo*, during the preparation of the main dish and rice and at the end to check the flavor
What do they use to measure salt?	Spoon, teaspoon and pinch of salt with fingers	Spoon and pinch of salt with a spoon	Plastic spoon and pinch of salt with fingers	Spoon and pinch of salt with fingers
Why do they use artificial seasonings?	N/A; not using artificial seasonings	To enhance flavor	To enhance flavor and save time	To enhance flavor
Who decides what to cook at home?	Herself. Sometimes she consults with her husband.	Herself. On the weekends, she consults with her husband.	Herself	Herself

## Data Availability

The data are subject to third party restrictions by the project funders.

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
