# Peer review of "Applying Customer Journey Mapping in Social Marketing to Understand Salt-Related Behaviors in Cooking. A Case Study"

_ijerph, 2021, doi:10.3390/ijerph182413262_

Round 1

Reviewer 1 Report

This is a generally well written paper but limited in sample size (n = 4), limited to one meal type (lunch), and does not sufficiently extrapolate from the results to reporting practical, concrete learnings for social marketing programs to reduce salt consumption. In effect, it is simply 'phase 1' of formative research to inform the development, pre-testing and implementation of such a social marketing campaign. I would suggest the authors expand their formative research before submitting for publication.  

I would also suggest that describing the study as "A Multiple Case Study" could be seen as misleading in that is simply a study of four subjects performing the same task. Also, the phrase in the abstract "yet it has been under utilized in social marketing" seems somewhat inconsistent with the referencing of several social marketing publications using the technique. Perhaps rephrase to something like "and could be useful in social marketing also" - or just omit that phrase. 

Reviewer 2 Report

This is a very interesting paper and potentially useful to the social marketing community as it highlights yet another example of using the customer journey approach to addressing a health issue.

There are areas that need highlighting/addressing more clearly such as:

There is a need to highlight the issue of potential effect of 'selection/treatment bias' given the fact that the subjects knew they were being 'observed'. Were the subjects made aware about the purpose of the research health focus and/or interest in salt intake? If not, then this should be made clear.

The suggestion regarding using spices to minimise the amount of salt could have been probed further through direct questioning, even as a follow up query. There is a a tacit suggestion that more salt intake is more to do with 'habits' rather than 'cultural taste' which can be misleading. 

Finally, language tidying up such as in line 144 talks about 'his son' and then 'her son'.

Reviewer 3 Report

The article addresses the issue of salt consumption in the kitchen of the Peruvian population. Too much salt is the cause of many diseases, mainly cardiovascular, high blood pressure.

The introduction is well presented.

The customer mapping method is applied as a research tool for the salt related cooking behavior of four women.

It is not very clear how are used the results of this research as base of marketing strategies in the health sector.

Some general comments refer to the way the research is organized and the usefulness of the results.

This mapping method in the kitchen and the identified touchpoints represents the food recipe. The four women would have been able to record their cooking process. They have to use the same main quantities into the same kind of pot, with the same type of teaspoon for adding the same ingredients for doing the same type of food.

In this way their results would have been comparable. The quantities of salt used could be measured for this purpose by looking at the records. 

A larger number of observations would have provided more information about salt consumption and segmentation variables about mother's age, number of family members, father's occupation, level of education, average income, rural - urban milieu, religion etc. The characteristics of the identified segments would be the basis for marketing strategies focused on these segments of population related to their different salt consumption.

The adding of natural or artificial seasonings could have been an objective to be studied.

Usually the women are cooking in a family and such a question is redundant. 

Specific comments (in lines):

  • The lines 102-104 about reward of 20$ value - It is not important either the value nor the existence of a reward.
  • Reformulate the lines 105-112, without the authors' names, just describe shortly the activities.
  • Also for line 123 - no author name within the text; at Contribution in the end of article you may specify for all.
  • Section 3 is too small as dimension; it can be at the end of section 2, where you describe the methodology. Simplify the phrase within Ethics, without the name of the ethics committee and the University name....
  • The description of participants may be presented in a table, as the structure of family, number of family members, occupation ... - more structured.
  • Line 223 - without author's name.
  • Lines 228-229 - the content is not important for the scientific paper.
  • Line 360 - without names of authors.

Round 2

Reviewer 1 Report

The authors have made several desirable changes but the implications for a salt reduction social marketing campaign are still very limited and the basic limitation of n = 4 remains. It is also an exaggeration to present 'customer journey mapping' as a 'new' or 'innovative' technique. In my view it is simply an observational technique - and such have been used in marketing research since the beginnings of marketing (eg see Crisp, Marketing Research, McGraw-Hill, 1957). 

Reviewer 3 Report

I appreciate the changes you have done and your answers in the cover letter.

I recommend you to write a scientific article based on the quantitative marketing research.

I think that you could use other qualitative method of research on this subject!

Some conclusions already known and expected are not worth the effort to find them!

Really I am not very excited by this article, but I recommend it to be published because I want you be encouraged to continue.

The question about average income may be formulated using intervals of income! It may be important to find if the groups of income influence the behavior of salt consumption....

Success!
